# The Function and Alteration of Immunological Properties in Human Milk of Obese Mothers

**DOI:** 10.3390/nu11061284

**Published:** 2019-06-06

**Authors:** Ummu D. Erliana, Alyce D. Fly

**Affiliations:** Indiana University Bloomington School of Public Health, Bloomington, IN 47405, USA; afly@indiana.edu

**Keywords:** maternal obesity, gestational weight gain, immunological properties, human milk, nutrition, health

## Abstract

Maternal obesity is associated with metabolic changes in mothers and higher risk of obesity in the offspring. Obesity in breastfeeding mothers appears to influence human milk production as well as the quality of human milk. Maternal obesity is associated with alteration of immunological factors concentrations in the human milk, such as C-reactive protein (CRP), leptin, IL-6, insulin, TNF-Alpha, ghrelin, adiponectin, and obestatin. Human milk is considered a first choice for infant nutrition due to the complete profile of macro nutrients, micro nutrients, and immunological properties. It is essential to understand how maternal obesity influences immunological properties of human milk because alterations could impact the nutrition status and health of the infant. This review summarizes the literature regarding the impact of maternal obesity on the concentration of particular immunological properties in the human milk.

## 1. Introduction

In 2014, the number of pregnant women with overweight and obesity were estimated at 38.9 million and 14.6 million, respectively, worldwide [1]. The prevalence of women of childbearing age with obesity in the US from 1976 to 2014 increased four-fold, from 7.4% to 27.5% [2,3]. In addition, the prevalence of pre-pregnancy obesity of mothers in the US was more than 20% [4,5]. According to 38 jurisdictions of the District of Columbia, New York City, and 48 states in the United States, the prevalence of mothers that had a normal body weight before pregnancy decreased from 47.3% to 45.1% from 2011 to 2015 [6]. A review by Fields et al. (2016) brought attention to the association between several bioactive components of human milk and adiposity in infancy [7]. There is great interest in understanding the contributions of maternal obesity to changes in the composition or functional properties of human milk. Obesity in lactating mothers was found to be related to changes in the concentration of several bioactive components of their milk [8]. For example, an increased leptin concentration found in human milk is noteworthy because leptin may contribute to maternal obesity [9,10,11,12,13,14]. Alterations of immunological constituents may influence the genetic, metabolic, and epigenetic processes in the child [15]. The excessive weight gain of an obese child was found to be related the increase in leptin and adiponectin concentrations in the milk from obese mothers [16,17]. Furthermore, increased insulin-like growth factor 1 (IGF-I) and ghrelin in human milk were also correlated with the increased growth rate of an obese infant [16,17,18,19]. 

Human milk consists of nutrients and active factors. In addition to nutritive functions, some milk constituents also have bioactive properties such as whey proteins (immunoglobulin, lactoferrin, and alpha-lactalbumin) and casein proteins (κ-casein and β-casein) [20]. Other bioactive proteins in human milk include immunological factors such as antibodies, live cells, cytokines or signaling molecules; enzymes such as lactoferrin, lysozyme, and bile salt stimulated lipase; glycoproteins or oligosaccharides (oligosaccharide-enriched fraction along with secretory IgA), glycolipids, and high molecular weight protein), alpha-lactalbumin, gut microflora like prebiotic, haptocorrin (vitamin B12-binding protein), and nutrients for the infants’ immune system [20]. The activities of these bioactive proteins need further study. 

Moreover, human milk composition, both over a single feeding and over the duration of lactation is unique and dynamic. Colostrum is the first milk produced (30 mL/24 h) from 30 to 40 h until a few days postpartum [21,22]. Transitional milk is the milk that is produced from 5 days to 2 weeks postpartum and mature milk is the milk that is produced after 2 weeks postpartum [22]. Mature milk has two types, foremilk and hindmilk. Foremilk is the initial milk of a feeding, while hindmilk is the last milk of feeding (which contains milk fat up to three-fold more than in foremilk [23]. Some characteristics of human milk depend on a multitude of factors of the mother. The DARLING (Davis Area Research on Lactation, Infant Nutrition and Growth) study showed important determinants of human milk composition to include maternal ideal body weight (%IBW), protein intake, nursing frequency, menstruation, and parity [24]. The composition of protein in human milk is also significantly influenced by weaning because weaning was shown to decrease the volume of milk production [25]. Furthermore, this study also showed that milk volume and protein level were inversely associated; if the milk production volume was decreased from >500 to <300 mL, the protein concentration would be increased from 1.23 to 2.01 g/100 mL [25]. The influence of obesity on the quality and quantity of human milk has been demonstrated in several studies. A significantly different and less diverse microbiome in human milk has been demonstrated over lactation period, maternal body mass index (BMI), and delivery mode [26]. The ratio of omega-6 to omega-3 of obese breastfeeding mother is increased, while the concentration of fatty acids (DHA, EPA, DPA,) and carotenoids (lutein) are decreased [15]. Alterations of bioactive properties in the human milk of obese mothers could increase the incidence of obesity, insulin resistance, type 2 diabetes, and other adverse metabolic outcomes [27]. 

The primary aim of this review is to describe the immunological functions of specific factors in human milk and the alterations that occur in conditions of overweight and obesity mothers. The secondary aim is to explore the other influential factors that stimulate alterations of specific immunological properties that also could influence their functions in human milk. 

## 2. The Function and Alteration of Immunological Properties in Human Milk 

Breast milk contains essential bioactive components that have been demonstrated by numerous studies. Many components studied have been found to provide a function, for example to prevent infections, heal diseases, and improve health status [28]. Many bioactive components still remain that are not well studied. Studies are needed to understand the implications of the alteration in the specific bioactive components to maternal and infant health. One study demonstrated that the alteration of specific bioactive components in human milk would influence infant health outcome in both the short term and long term [29]. The understanding of this dynamic variation of human milk is needed because it may suggest practices needed for breastfeeding management and distribution of milk donations [22]. Modification of dietary intake and immunization can optimize the concentration of bioactive components in human milk [30,31]. Table 1 lists the bioactive components of human milk, and whether studies have reported any alterations in the concentration of milk of obese mothers. The functions of these bioactives and details about the changes are described further in the text.

### 2.1. Anti-Microbial

Lactoferrin is one of the major iron-binding glycoproteins in colostrum that is immunomodulatory [49]. Lactoferrin is bound to iron and helps in uptake of iron in the cell by specific receptors in the infant intestinal tract [50]. Lactoferrin also functions at the level of DNA, as a transcription factor that influences for example, cell signaling proteins and immune protein synthesis [20,51]. Lactoferrin acts to constrain bacterial growth by binding iron, limiting iron needed for growth of bacteria and other pathogens [52]. The formation of lactoferrin without iron has also been found in human milk. This form was demonstrated to eradicate *Candida albicans*, *Escherichia coli*, *Pseudomonas aeruginosa*, *Streptococcus mutans*, *Streptococcus pneumoniae*, and *Vibrio cholerae* [53]. Lactoferrin concentration was significantly greater in colostrum of mothers who were greater than the 90% Weight for Height, an older surrogate measure for obesity (that corresponds to a BMI > 30 kg/m^2^) [32]. 

Lactadherin is a glycoprotein associated with milk fat globules of human milk together with mucins, xanthine oxidase, and butyrophylin [20]. The main role of lactadherin may be to protect the newborn infant from rotaviral infection, a common cause of diarrheal disease and gastroenteritis [54]. Lactadherin works against the infection by creating apoptosis in infected cells of the infant and decreases inflammation by inhibition of TLR4 and the NF-κB signaling cascade [55,56,57]. Lactadherin is not digested in the stomach and passes to the intestine to maintain gut health by ameliorating inflammation [57,58,59]. A study in Mexico with 200 infants demonstrated that infants who were breastfed by milk that contained a low concentration of lactadherin developed severe diarrhea [54]. Other infants who received high levels lactadherin from human milk were asymptomatic of diarrhea. 

The catalytic reactions of lactoperoxidase have bactericidal effects that kill Gram-positive and Gram-negative bacteria [60,61]. This bactericidal effect results from catalytic oxidation of substrate such as thiocyanates with hydrogen peroxide resulting in hypothiocyanite ion (OSCN^−^). The concentration of lactoperoxidase of human milk at the first 6 months was found to be constant between 1–1.5 units/mL [62]. Lactoperoxidase in human milk was shown to detoxify H_2_O_2_ both in the infant gut and mammary gland of mothers, with additional anti-microbial functions. Whether lactoperoxidase concentration in the human milk of obese mothers is altered has not yet been determined.

Another antimicrobial constituent of milk, lysozyme, is present at a concentration that is 3000-fold higher in human milk than cow’s milk. It is an active enzyme that works together with lactoferrin to effectively kill Gram-negative bacteria [63]. A study of the concentration of lysozyme in human milk of obese mothers was not found. 

Mucins, another antimicrobial factor, make up one of three major protein fractions of human milk, along with casein and whey [64]. Mucins are a type of glycoprotein that consist of up to 80% carbohydrate, including mannose, as well as sulfonic acid [65]. MUC1 and MUC4 were identified in human milk for the first time by Liu et al. (2012) [66]. Furthermore, their study demonstrated that MUC1 was better than MUC4 in protecting the human epithelium cells (FHs 74 Int cells, CaCo-2 cells) from invasion by Salmonella [66]. Mucins are part of a passive immunity in human milk that protect the infant small intestine and stomach by inhibiting the binding of pathogens [67,68]. There was no study found that compared mucins concentration in the human milk of obese mothers to mothers with normal BMI.

### 2.2. Cells

The existence of lymphocytes in human milk was first discovered in colostrum [69]. Another study also showed that GFP+ leukocytes were transferred from mothers to their infants through breast milk [70]. T lymphocyte cells (CD8+, CD4+, and CD19+) are produced by GFP+ leukocytes in the Peyer’s patches (PPs) [71]. Moreover, Th-2 lymphocytes were shown to contribute to produce specific cytokines such as IL-4, IL-13, IL-15 [56]. This study also demonstrated that the composition of lymphocytes in human milk and blood were different [72]. There were no studies found that compared lymphocyte concentrations in milk of obese mothers with other mothers. 

Macrophages and the mammary endothelium support the production of TNF-Alpha in human milk [73,74]. Macrophages protect the infant from the infection by pathogens by activation of T-cells [75,76]. The cells move into the maternal blood and then are shifted to human milk by the mammary epithelial cells [22]. There were no studies found that reported comparisons of macrophage concentration in milk between women with obesity and normal weight conditions.

Neutrophils are another type of leukocyte that are abundant in the colostrum [77]. The three activities of neutrophils in human milk are bactericidal, phagocytic, and enzymatic [78]. Microbiocidal activity occurs through the production of oxidants and granule enzyme activities. In addition, neutrophils demonstrated significant bactericidal activity against *Staphylococcus aureus* [78]. This study by Grazioso and Buescher (1996) also demonstrated the effective ratio for phagocytosis between neutrophils and staphylococci cells at 2:1. A study by Islam et al. (2006) demonstrated that the percentages of breast milk phagocytes ( neutrophil-macrophages) of a cell count in mothers with underweight, normal BMI, and overweight were 46.6% ± 19.35, 60.24% ± 6.93, and 55.55% ± 16.16, respectively [33]. 

Stem cells have been found in human milk that differentiate into luminal, basal, and myoepithelial layers [79]. These stem cells can regulate an octamer-binding transcription factor 4 expression that repairs human cells. The stem cells in human milk protect the cells of the infant [80]. The stem cells in breast milk were hypothesized to function as a neuroprotective factor in a study that used intra-nasal application of breast milk for rescuing preterm neonates from brain injury and to improve long-term neurocognitive improvement of preterm infants [81]. A review by Kakulas (2015) concluded that the number of lactocytes (mature mammary epithelial cells) in obese mothers was lower than other reports of lactocytes in healthy mature milk likely due to lower milk supply of obese mothers. In addition, a study by Twigger et al. (2015) in a review by Kakulas (2015) explained that the number of lactocytes in mothers with obesity was decreased because of the lower expression CK18 expression. Moreover, the lower expression of CK18 expression (a marker of luminal epithelial cell activity such as lactocytes) would reduce epithelial tissue capability to synthesize milk in mammary glands in obese mothers [34,80].

### 2.3. Chemokines

Chemokines are a type of cytokine that may stimulate cell movement [22]. Chemokine production is influenced by lactoferrin and a derivative of lactoferrin [82]. These factors have not been studied with regard to maternal body weight and lactation.

G-CSF functions as a hematopoietic growth factor, to stimulate the proliferation of clonal and neutrophil progenitor differentiation [83]. A study by Wallace et al. (1997) showed that the concentration of G-CSF in 30 samples of mothers with normal BMI ranged from 14 to >2500 pg/mL [84]. Again, there was no information found as to how high maternal body weight affected this factor.

The production of macrophage migration inhibitory factor (MIF) is equivocal with some reports that MIF is not produced by specific human cells [85]. Another study demonstrated that MIF is produced by a large number of T-cells [86]. The specific interleukins found to induce MIF production are tumor necrosis factor-alpha (TNF-α) and interferon gamma (IFNγ) [85,87]. There was no information found regarding how or whether MIF is affected by maternal body weight.

### 2.4. Cytokines

Cytokines interact with the other cells and cross the intestinal barrier to enhance or defend, and reduce inflammation [22]. The cytokines in human milk have roles as immunomodulatory and anti-inflammatory to reduce infection in the infant [88]. Additionally, mammary glands need cytokines for growth and development. Cytokines regulate proliferation and inflammation of cells [89].

Both regulation and development of mammary gland function are mediated by IL-6 and TNF-Alpha [90]. In addition, fever and systemic inflammation also correlate with IL-6, for example IL-6 was higher when mastitis occurred in the lobes [91,92]. IL-6, IL-10, and TGF-β are maternal cytokines associated with maturation of the intestinal immune system. These cytokines also regulate IgA production, development, and differentiation of cells [93]. A study by Whitaker et al. (2017) found no significant association between the maternal weight and IL-6 level [8]. Another study by Fields et al. (2017) also demonstrated similar results of no association between BMI of maternal mothers and IL-6 concentration in their milk [18]. However, a study by Collado (2012) showed contrasting results. IL-6 concentration was higher in colostrum of overweight and obese mothers (81.85 pg/mL) than in colostrum of healthy BMI mothers (62.86 pg/mL), while IL-6 concentrations in 1-month milk of overweight and obese mothers (13.22 pg/mL) was lower than IL-6 concentration of normal BMI mothers (22.12 pg/mL) [35].

The main functions of IL-7 are to support lymphoid development, produce T-cells in the thymus, and assist in T-cell survival at secondary of peripheral lymphoid tissue [94]. IL-7 from human milk is absorbed by the infant through the intestine.

In a study of mothers with varying BMI, the concentrations of TNFα in colostrum and 1-month milk of mothers with normal BMI were reported at 9.87 and 10.60 pg/mL, respectively [35]. However, these values differ greatly from the study by Meki et al. (2003) [91]. This study concluded that TNFα is a proinflammatory cytokine that increased over the course of lactation; however, the values seem to be similar with no standard deviation indicated. These authors also reported that the TNFα concentration in colostrum and 1-month milk of mothers with BMI ≤25 kg/m^2^ and BMI >25 kg/m^2^ were 11.41 and 10.23 pg/mL, respectively. However, there was no statistically significant difference in the TNFα content of the milk in these mothers [35]. The values for this cytokine are logarithmically different than those reported by others in the literature [35]. Therefore, while no differences were detected by Meki et al. the inconsistency with the literature makes it hard to conclude with certainty that the concentration of TNFα is not influenced by maternal BMI.

### 2.5. Growth Factors 

Epidermal growth factor (EGF) is a growth modulator that can be found in amniotic fluid [95]. The function of EGF in human milk, as a growth and weight regulator, is related to appetite of the infant [19]. Another role of EGF is to support the maturation of infant gut by development, from a high intestinal permeability to become more selectively permeable [95]. A study by Khodabakhshi (2015) demonstrated that the EGF concentration in milk of obese mothers (0.038 ng/mL) was significantly lower (*p* = 0.013) than that of mothers with normal BMI (0.040 ng/mL). Furthermore, low concentration of EGF in the mother’s milk was significantly associated with higher infant body weight [19].

HB-EGF is a growth factor that can be found in both amniotic fluid and human milk [96,97]. HB-EGF appears to protect the infant intestine, specifically the intestinal epithelium from cytokine-induced apoptosis and hypoxic necrosis by decreasing the level of nitrogen and reactive oxygen species produced [98,99,100,101].

The Insulin-like Growth Factors (IGF)-I in human blood were found in 1968 by Jacobs et al. [102]. IGF-1 has an important role in infant growth where the high concentration of IGF-1 in the human milk promotes rapid growth to the infants that may be associated with overweight and obesity in later life [103]. IGF-1 in breast milk is thought to protect premature infants from retinopathy of prematurity (ROP) in the first 6 weeks of life; IGF-1 deficiency during the first weeks of neonatal life was associated with ROP [104]. In addition, infants who consumed only breast milk had a lower incidence of ROP than the infants who consume infant formula [104]. The concentration of IGF-1 in the human milk of obese mothers (89.63 ng/mL) and normal mothers (75.09) was not significantly different (*p* = 0.787). 

Specific neurons of the peripheral nervous system need nerve growth factor (NGF) to survive [105]. NGF has been shown to stimulate the phosphorylation of tyrosine on the TrkA protein receptor in human mast cells-1 (HMC-1) [106]. There was no literature found that described the effect of overweight or obesity in mothers on milk content or functions of NGF.

Human milk contains vascular endothelial growth factor (VEGF) and epidermal growth factor (EGF) at concentrations that are 100 times higher than blood of lactating mothers [107]. The concentration of VEGF from 33 samples of lactating mothers ranged from 12.6 to 155.0 (median 50.0) ng/mL. The concentration of VEGF was significantly positively correlated with the EGF concentration, but negatively correlated with hepatic growth factor (HGF) concentration [107]. VEGF receptors, such as fetal liver kinase 1, are expressed primarily in endothelial cells [108]. There were no studies found that investigated the effect of maternal overweight or obesity on concentrations of VEGF and HGF.

### 2.6. Hormones

The role of adiponectin is to regulate metabolism of the body, specifically affecting satiety, tissue sensitivity to insulin, stimulate glucose uptake, and decrease energy expenditure [109,110,111]. Studies by Martin et al. (2006) showed a positive correlation between maternal BMI and the concentration of adiponectin [36]. Furthermore, higher adiponectin in milk increases the risk of being overweight in childhood [109]. In contrast, another study specifically demonstrated lower weight gain and leaner body of neonates [112]. 

Calcitonin has functions to inhibit gastric acid production, or regulate fluid balance, food intake, and gastrointestinal motility [113,114,115]. The calcitonin concentration level in the human milk was found to be 10–40-fold greater than the concentration in the serum [116]. Lactating mothers have elevated serum calcitonin which is released into their milk [117]. The effects of maternal obesity on milk calcitonin have not been reported.

Ghrelin was identified by Kojima et al. (1999), as a 28-amino-acid-peptide. It was found to be predominantly expressed in the stomach [118], but it is also expressed in the pituitary, hypothalamus, and pancreas and functions to regulate food intake, sleep, behavior, gastric acid secretion and motility, glucose metabolism, exocrine and endocrine pancreatic function, cardiac performances and vascular resistance, energy metabolism, cell proliferation and survival, stimulates of ghrelin (GH), Arginine vasopressin (AVP), prolactin (PRL), adrenocorticotropic hormone (ACTH) secretion, and inhibits the gonadotropin secretion [119]. Ghrelin in the human milk functions as a growth factor that stimulates feeding by the infant [120]. A study by Khodabakhsh (2015) showed a negative correlation between obese mothers and the concentration of ghrelin in milk [19]. A similar result also was also demonstrated in a study by Zhang et al. (2011), where the concentration of ghrelin in human milk of normal weight mothers was higher than that in mothers with obesity [37].

Insulin in human milk functions as a hormone, and also influences the microbiome community in the lumen of the gastrointestinal tract [10]. Mothers with obesity accumulate more insulin in their human milk than mothers with normal body weight [109,121]. A study by Ley et al. (2012) specifically showed that the insulin concentrations were higher in the milk from obese mothers than mothers with normal BMI at 3 months post-partum [121]. The impact of higher insulin and glucose in human milk is more rapid growth of the infant which may increase risk for childhood obesity because of the higher energy content of diabetic breast milk than breast milk of healthy mothers [122,123]. In addition, Plagemann et al. (2002) observed a positive correlation between infant overweight and volume of diabetic breast milk (DBM) consumed at 2 years of age compared to infants who consumed banked breast milk from non-diabetic mothers [124]. This study showed early neonatal consumption of breastmilk from diabetic mothers was positively correlated to risk of overweight, and impaired glucose tolerance (IGT) during childhood [124].

Leptin is an adipocyte-derived hormone that was first found in 1994 [125]. Opposing the adiponectin function, leptin functions to increase appetite and suppress energy expenditure. It is considered a regulator of long-term energy balance [126]. The mean concentration of leptin at 1 month postpartum in the human milk of 50 normal weight mothers was 2.5 ± 1.5 ng/mL [9]. The concentration of milk leptin significantly decreased, up to 33.7% after 6 months of lactation, this may be in response to the decrease in fat mass of the mothers [18]. A study by Fields et al. (2017) demonstrated other determinants of leptin concentration in human milk such as BMI category, and gender of infant [18]. This study also showed a significant positive correlation between maternal BMI and leptin concentration in human milk. Furthermore, another study demonstrated higher concentrations of leptin in the milk of obese mothers, up to three times the concentration in milk of normal weight mothers [10]. Uysal et al. (2002) also found a significant correlation between high leptin concentration and maternal obesity [11]. The positive association between maternal obesity and leptin concentration in milk also noted by Andreas et al. (2014) and Quinn et al. (2014) [12,13]. At the 3rd and 28th day after delivery, the leptin concentration in milk of mothers who had BMI over 25 showed a significantly higher concentration than the milk of normal weight mothers [14]. Other data supporting this finding, are that increasing concentrations of leptin were found in milk of obese mothers at 1 month postpartum (4.8 ± 2.7 ng/mL) [9]. 

Obestatin is a hormone that was found in 2005 in the gastrointestinal tract [127]. Obestatin in human milk may control the appetite and gastrointestinal function of infant as the infant adapts to receiving milk [38,128]. The concentration of obestatin is negatively correlated with the BMI of mothers [38]. Therefore, the concentration of obestatin in the milk of overweight and obese mothers will be lower than the concentration of obestatin in milk of normal weight mothers. 

Resistin is a hormone in breast milk that regulates fetal growth, appetite, and metabolic development of the infant [129]. The median concentrations of resistin in 23 samples of breast milk were 0.18 ng/mL (IR = 0.44) [40]. Study by Savino et al. (2012) and Andreas at al. (2016) found that the concentration of resistin was not altered in milk of mothers with obese BMI [39,40]. However, the concentration of resistin was higher in the serum of mothers with obese BMI [39]. The higher concentration of resistin in the serum and milk would increase the concentration of hormones (cortisol, estradiol, leptin, progesterone, prolactin, triiodothyronine, and thyroxine) and inflammatory marker (C-reactive protein) [130]. 

Somatostatin functions to inhibit gastrin secretion [131]. The highest concentration of somatostatin was found on the first day postpartum, in colostrum [132]. The concentration of milk somatostatin is 7.2 times higher than the concentration in the plasma of mothers, due to an active transport process moving somatostatin from blood to mammary gland. Reports of maternal obesity on somatostatin concentration have not been found.

### 2.7. Immunoglobulins

The major immunoglobulin found in human milk is Secretory IgA (sIgA, 80%–90%). This antibody is transferred from the milk to the infant, at the rate of 0.3 g/kg/day. About 10% is absorbed from the intestines and enters the infant blood [133]. A deficiency of sIgA in the mucous membranes can be substituted by a high concentration of IgM antibodies, particularly for infants with selective IgA deficiency [134]. Carbonare (1997) showed that sIgA fully protects the intestinal tract from enteropathogenic *E. coli* antigens, and is not destroyed by digestion [135]. A study by Islam et al. (2006) also compared the sIgA concentrations among three BMI groups. The study showed that IgA concentration in the human milk of overweight (5.60 ± 1.47 g/L) and normal mothers (5.67 ± 165 g/L) were higher than IgA concentration in malnourished mothers (5.22 ± 1.68 g/L) [63]. The standard deviation of the IgA concentration in the milk of the mothers with normal BMI may be a typographical error as it seems very high.

Unlike IgA, IgG and IgM are completely digested in the small intestine [20]. Thus, intact IgG is not present in significant quantities at the intestinal mucosal surface of infants. IgG is present at a low concentration (0.1 mg/mL) in human milk though it has a role in activating the complement system and antibody-dependent cytotoxicity [136]. The concentrations of IgG in breast milk of mothers with underweight, normal BMI, and overweight were 0.095 ± 024 g/L, 0.096 ± 024 g/L, and 0.093 ± 020 g/L, respectively [33]. 

IgM concentration in the colostrum of human milk is the second highest of the immunoglobulins. The mean concentration of IgM is 0.47 ± 0.10 g/L [33]. IgM protects the infant intestinal mucosal surface from viruses and bacteria [136]. Islam et al. (2006) demonstrated the mean concentration of overweight mothers and normal mothers were 0.47 ± 0.01 g/L and 0.47 ± 0.09 g/L, respectively. 

Insulin-like growth factor-binding proteins can be found in human milk and include insulin growth factor I (IGF-1) and insulin growth factor 2 (IGF-2). Colostrum contains the highest concentration of IGF compared to the later milks [137]. IGF-1 functions to protect the enterocytes in the intestine from damage by oxidative stress, stimulates erythropoiesis, and is related to increasing hematocrit [137]. IGF-binding protein 2 is higher in preterm milk than term milk [138]. IGF-1 concentration is influenced by BMI of mothers whereby IGF-1 concentration was found to decrease significantly in the obese breastfeeding mothers [19]. 

### 2.8. Lipids

Lipids contribute up to 44% of the total energy in human milk [139]. Lipids can have an antimicrobial function in the infant intestine [140]. In addition, the activity and development of antimicrobial function in infant’s gut is supported by milk fats through provision of nutrients [141]. Cell membranes of bacteria were shown to be damaged by antimicrobial activity of free fatty acids and monoglycerides [142]. A study by Fujimori et al. (2015) demonstrated that both the triglyceride and cholesterol concentration in the colostrum of overweight and obese mothers were higher than concentrations in colostrum of normal weight mothers [42]. Another study by Makela et al. (2013) showed that the fatty acid composition in human milk of overweight mothers was significantly higher in total saturated fatty acids and lower in n-3 fatty acids than in milk of normal weight mothers [43]. Furthermore, unsaturated to saturated fatty acid ratios in milk and the ratio of n-6 to n-3 was higher in overweight mothers versus normal weight mothers. A complete analysis of the various in fatty acids concentration in obese and normal mothers has been conducted by Panagos et al. (2016) [15]. A meta-analysis and a randomized controlled trial (RCT) showed associations between fatty acid composition in human milk and child growth were not significant [143,144,145,146].

### 2.9. Microbiota

Human milk supports proliferation of the milk microbiota by regulating the balance of specific microbiota such as, *Lactobacillus, Bacteroides,* and *Bifidobacteria*. The proliferation of these microbiota stimulates the activation of T regulatory cells and transform intrauterine TH2 predominant to rebalance TH1/TH2 [147]. About 200 kinds of important bacteria grow to constitute the first gut microbiota in the newborn [148]. *Bifidobacterium* and *Lactobacillus* are the primary probiotic in the gut that provide nutrients to the newborn intestine and protect it by establishing an acidic environment rich in short chain fatty acids (SCFAs) [148,149]. Dendritic cells (DCs) and CD18+ cells function to capture the microorganism from the mother’s intestinal bacteria and translocate them to lactating mammary glands, therefore the number of DCs and CD18+ cells will increase late in pregnancy and during lactation [150]. A decreasing number of *Bifidobacterium* and increasing number of *Staphylococcus* have been demonstrated in human milk of obese and overweight mothers [35,44]. Furthermore, the higher concentration of *Lactobacillus* and the lower of concentration of *Bifidobacterium* for 3 months may contribute to increased risk of overweight and obesity to the infant [151]. Another study also demonstrated that obese infant and obese children have significantly different microbial flora because of different colonization through breast milk bacteria [152]. The alteration of the microbiota is not the only influence on nutritional status of the infant, but may be related to increased risk of asthma, allergies, inflammatory bowel disease, and type 1 diabetes. Thus, by Turnbaugh et al. (2012), Dibaise et al. (2009), Karvonen et al. (2014), Kostic et al. (2015), Frank et al. (2007) and Cabrera-Rubio et al. (2012) concluded that the bacteria in human milk of obese mothers may influence the health status of their infants [153,154,155,156,157,158]. 

### 2.10. Nucleic Acids

The degradation of milk nucleic acid polymers (RNA) provides nucleotides, which have been found to improve immune function, enhance the bioavailability of iron, adjust the microflora in the intestine, alter plasma lipids, and support growth and maturation of the gut [159,160,161,162,163]. Production of nucleotides in human milk needs further study because there is no study yet that compares the concentration of nucleotides in the milk of mothers with obese and normal BMI. 

### 2.11. Oligosaccharides and Glycans

Human milk oligosaccharides (HMOs) protect infants from microbial infections, protect gut microbiota, prevent microbial adhesion, and invasion in the intestinal mucosa [164]. HMOs are small carbohydrates that bind pathogen bacteria and also facilitate the establishment of a protective microbiome in the intestinal tract of the infant [165]. HMOs are the main glycan and influential in the human milk because they inhibit the growth of pathogenic bacteria by binding to the pathogen and preventing pathogen binding to the intestinal epithelium [165]. HMOs prevent the attachment of viruses and bacteria surface on intestinal epithelium [166]. HMO is involved in development of the submucosa lymphoid structure and intestinal epithelial cell (IEC) barrier [166]. Colostrum contains two kinds of HMOs, 2′-fucosyllactose and 3′-galacto-syllactose, that reduce cytokine production and the inflammatory reactions [167,168]. A study by Azad et al. (2018) showed no significant correlation between the maternal BMI and the concentration of HMO in human milk [45]. This is of interest because a variation of HMO composition in human milk could influence the gut microbiome of the infant and variation in HMO has been shown to be significantly related to growth and body composition of the infant [169]. 

Prebiotics such as glycosaminoglycans, oligosaccharides, glycoprotein, and glycolipids are needed to support growth of the microbiota in the intestinal tract [166]. Glycosaminoglycans may be protective to infants, particularly the newborn, because the number of these polysaccharides are highest at the first month of lactation [170]. Furthermore, a study by Coppa et al. (2012) also demonstrated that the number of glycosaminoglycans in preterm human milk was higher than in term milk. A study of the association between BMI mothers and metabolic capacity of glycosaminoglycans was shown by Cardo et al. (2018). They found significant degradation of glycosaminoglycan activity in human milk of obese mothers compared to normal BMI mothers [46].

The concentration of soluble CD14 (sCD14) in the colostrum is 20-fold greater than the concentration in maternal serum [171]. A function of sCD14 in the infant intestine is to modulate the innate and adaptive immune response of bacterial colonization [171]. A study by Collado (2012) found that sCD14 in colostrum (28.22 μg/mL) and 1-month milk (5.54 μg/mL) of normal BMI mothers was higher than that of overweight and obese mothers, which contained 23.21 and 4.35 μg/mL, respectively [35]. These differences were not significant.

Osteoprotegerin is part of the TNF super family components that inhibit the activation of TNF-induction to prevent proliferation process of T cells. The concentration of osteoprotegerin in epithelial cells of the human mammary gland and human milk is 1000 times higher than the concentration in human serum [172]. Osteoprotegerin function in the Th1 cells may re-balance the concentration of Th1 and Th2 of newborn infants [173]. However, there was no study found that investigated maternal obesity and overweight on osteoprotegerin concentration in human milk.

### 2.12. Other Proteins

Alpha-lactalbumin of human milk is a bioactive protein that can be found in the whey fraction of milk [174]. Alpha-lactalbumin is present in both cow milk and human milk, and contains 123 amino acids [175]. Alpha-lactalbumin makes up 22% of total protein and 36% of whey protein in human milk [176,177]. The alpha-lactalbumin functions in synthesis of lactose and regulation of water circulation to human milk through osmotic systems [176]. The level of alpha-lactalbumin in milk is influenced by genetic, environment, and dietary factors [178]. For these reasons, alpha-lactalbumin plays a vital role as a bioactive protein. A previous study showed that the concentration of alpha-lactalbumin in human milk was influenced by nutritional status of mothers. Standard weight (SW) of mothers significantly influenced the concentration of alpha-lactalbumin in human milk [47]. Specifically, the concentration of alpha-lactalbumin in human milk of mothers who had SW, <10% SW, 10%–25% SW, and >25% SW were significantly different at 16–30 days and 31–60 days after delivery. 

α1-Antitrypsin is a protease inhibitor in human milk. It has been shown to be an influential protein together with antichymotrypsin [179]. α1-Antitrypsin in human milk can be detected in the stool of human milk-fed infants [180,181]. The function of α1-antitrypsin and antichymotrypsin may be to permit absorption of certain bioactive proteins by limiting the digestibility of certain proteins in the infant gut [182]. Furthermore, α1-antitrypsin and antichymotrypsin affected the total nitrogen balance of infants who are fed by human milk. The inhibitory activity of α1-antitrypsin was not affected by pH (down to pH 2) or temperature [183]. It may be relatedly a protector of bioactive proteins (such as lactoferrin) at the small intestine, particularly in the upper part [183,184]. There was no study found that explored maternal obesity or overweight and α1-Antitrypsin.

Alpha-amylase is another significant bioactive protein in human milk [185]. The newborn infant has low concentrations of salivary amylase and pancreatic amylase activity. Milk amylase therefore may function to compensate for the low infant alpha-amylase from human milk [186]. Future studies should be conducted to better understand the significance of milk alpha-amylase for utilization of carbohydrate in mixed diet-fed infants [182]. There was no report that explored alpha-amylase concentration in milk of overweight and obese mothers.

Bile Salt Stimulated Lipase (BSSL) has been identified in human milk but not cow milk, where it contributes to 1%–2% of total milk protein. BSSL is influential as an active enzyme to digest lipid in the lumen of the infant gut. It hydrolyzes the esters of vitamin A and cholesterol, lyso-phospholipids, and other milk fats such as monoglycerides, diglycerides, and triglycerides [20]. A study by Andersson et al. (2007) demonstrated that BSSL can be destroyed or deactivated by the pasteurization process. Therefore, preterm infants who consume pasteurized milk from donor mothers show reductions in lipid absorption. Anthropometric measurements of length, weight, heel-to-knee of infants who consumed non-pasteurized milk increased more in these growth markers than infants who consumed pasteurized milk [187].

Human milk proteins are often classified according the nature of how the colloids are distributed in milk, as either two components, whey or casein. Casein proteins are distinguished by their arrangement in micelles. Human milk contains 20%–40% caseins [20]. The concentration of β-casein is the highest among other human caseins. β-casein is a highly phosphorylated protein [188]. MUC2 functions to protect the human small intestine layer by multiple mechanisms [189]. The adhesion of *Helicobacter pylori* is prevented by κ-casein activity at the gastric mucosa. More specifically, κ-casein at the epithelial cell surface has role as a soluble receptor analogue which prevents the binding of mucosal epithelium to bacteria [182]. The ratio of the concentration of casein and whey decreases over time [190]. At the first 2 weeks after delivery, the ratio concentration of casein and whey was found to be 20:80. After 2 weeks postpartum the ratio concentration of casein and whey changed to 35:65 and stayed constant. The beta-casein expression was reduced in milk of overfed rats, which suggests a similar effect might occur on beta-casein concentration in the human milk of overweight and obese mothers [48]. 

C-reactive protein is an inflammatory protein marker that is primarily produced by the liver [191]. Maternal obesity was found to be associated with an alteration of CRP. A study by Whitaker et al. (2017) showed that the concentrations of CRP in the human milk of 126 obese and or mothers with excessive weight gain were significantly higher than mothers with the normal weight gain or lower weight. High concentrations of CRP level in infant serum can increase the risk of cardiovascular disease in the long term due to CRP level functions to control serum cholesterol level [192]. 

Folate-binding protein regulates the metabolism of the vitamin folic acid, through distribution, absorption, and retention [193]. The soluble folate-binding protein (FBP) function is a glycosylated compound that binds the vitamin in a way that prevents proteolytic degradation in the low gastric pH. The glycosylation appears to provide protection from digestion [194]. Furthermore, folate bound to FBP is gradually released in the small intestine mucosa [195]. There was no study found that reported effects of maternal obesity or overweight on FBP concentration in milk.

Haptocorrin is an anti-microbial substance that was found to reduce *E. coli* and is stable against proteolytic digestive enzymes [196]. In addition, haptocorrin remains undigested at pH 3.5 by pepsin and pancreatic enzymes [196]. High concentrations of haptocorrin are not needed for pathogenic activity. Another function of haptocorrin in human milk is as a substitute for intrinsic factor, that helps in the absorption of Vitamin B_12_ in the newborn (acting as a vitamin B_12_-binding protein) [197]. There was no study found that reported effects of maternal obesity or overweight on haptocorrin concentration in milk.

## 3. The Influence of Other Factors on the Levels of Immunological Properties in Human Milk

The composition of human milk is also influenced by other maternal factors, such as diet, age, ethnicity, metabolic health, type of delivery, smoking, amount of sleep, stress, and physical activity. Additionally, it is influenced by physiological factors such as diurnal variation, stage of lactation and stage of nursing, as well as factors related to the infant, including gender, birthweight, body composition, and behavioral factors, such as ad libitum feeding and time between feedings [7]. The main influential factors on human milk composition are preterm delivery and stage of lactation [198]. Studies that demonstrate an influence of other factors that affect immunological properties of milk are reported in Table 2.

## 4. Conclusions

Based on literature that were collected and reviewed, obesity and overweight of mothers appears have a clear influence on the concentration of 29 immunological properties in human milk. An alteration of any of these immunological properties could influence the function of the human milk. An increase or decrease of immunological properties in human milk are predicted to influence the health status of infants in both the short term and long term. Further research is needed to study the correlation of high BMI with the 31 yet unstudied or unreported immunological properties in human milk and the risk of specific diseases to the infants.

There are some limitations of this review. First, the range of years for publications of studies reviewed was not limited. Some current studies might have different results compared to older studies because of different number of participants, data collection methods, immunoassay methods and detection, and statistical analyses. Second, many studies did not classify the samples collected as to whether they were colostrum, hind milk, and fore milk. Differences in the composition of milk over the duration of lactation should have different concentration of immunological properties. Furthermore, these variations might influence the output of immunological actions related to their function as protectors and supporters for infant growth and development. 

## 5. Practice Points

Maternal overweight and obesity has been found to affect 29 immunological properties of human milk. While a complete understanding of the implications of these changes to the infant are not known, they add to the body of evidence supporting healthy weight management:
Medical practitioners, dietitian/nutritionists, and nurses should educate the pregnant mothers with BMI > 30 kg/m^2^ about the risk of obesity to mother and infant and suggest restricting excessive caloric intake outside of the recommended amounts needed for healthy infant growth. The health care professionals support the pregnant mother in obesity by providing weight management targets and the information about nutrient dense eating plans or menus.After delivery, exclusive breastfeeding can be promoted as another component of energy expenditure which will help prevent further weight gain by the obese mother. Breastfeeding mothers should not follow a weight loss program until 6–8 weeks after delivery because it may influence the quality and quantity of milk production. The overweight and obese mothers should be encouraged and supported to exclusively breastfeed for at least for 6 months.At 6–8 weeks postpartum, obese mothers should be encouraged to pursue healthy weight loss programs with nutrient dense eating plans after lactation is well established. Health professionals can emphasize that achieving a healthier BMI provides health benefits to both the infants (by normalizing the immunological properties concentration in milk) and to the mothers (by decreasing the risk of metabolic diseases, such as, heart disease, diabetes mellitus, and cancer).Even though composition of some immunological factors in milk changes in obesity, there is not enough evidence at this time to suggest that the changes in the milk should preclude breastfeeding the infant, as the benefits of breastfeeding are well established. In fact, obese mothers in the US, Europe, Brazil, Latin America, Singapore, India, South Africa, and Australia, that cannot breastfeed, can consult with international board certified lactation consultants (IBCLC) and pediatricians to consider obtaining donor milk from a milk bank in one of these countries. 

## 6. Research Agenda

The association of maternal obesity with immunological properties reported as unstudied in this review need investigation.The alteration of specific immunological properties and these effects on infant health condition (when breastfed by obese mothers) should be examined both in the short term and long term. A reduction of protective factors could have a negative impact on infant health. The differences in infant health status after exclusive breastfeeding by obese mothers should also be explored for obese mothers that practice nonexclusive breastfeeding. Exclusive breastfeeding has been widely encouraged due to the benefits for mothers and infants. It is conceivable that alterations in the immunological properties due to obesity may affect the usual advantages associated with exclusive breastfeeding.

## Figures and Tables

**Table 1 nutrients-11-01284-t001:** Alteration of immunological properties in human milk of obese mothers.

No	Bioactive Components	Alterations	References
1.	Antimicrobial		
	a. Lactoferrin	Increase	Houghton et al., 1985 [32]
	b. Lactadherin	None reported	-
	c. Lactoperoxidase	None reported	-
	d. Lysozyme	None reported	-
	e. Mucins (MUC1 and MUC4)	None reported	-
2.	Cells		
	a. Lymphocytes	None reported	-
	b. Macrophages	None reported	-
	c. Neutrophils	Increase	Islam et al., 2006 [33]
	d. Stem cells	Decrease	Twigger et al., 2015 [34]
3.	Chemokines		
	a. Granulocyte Colony Stimulating Factor	None reported	-
	b. Macrophage migration inhibitory factor (MIF)	None reported	-
	c. Chemokine receptors (CXCR1/CXCR2)	None reported	-
	d. CXCL-9 (MIIP)	None reported	-
4.	Cytokines		
	a. Interleukin-1 beta (IL-1β)	None reported	-
	b. Interleukin-2 (IL-2)	Increase	Collado et al., 2012 [35]
	c. Interleukin-4 (IL-4)	Increase (colostrum) Decrease (1-month milk)	Collado et al., 2012 [35]
	d. Interleukin-6 (IL-6)	No alteration	Whitaker et al., 2017 [8]
		Increase	Collado et al., 2012 [35]
	e. Interleukin-7 (IL-7)	None reported	-
	f. Interleukin-8 (IL-8)	None reported	-
	g. Interleukin-10 (IL-10)	Increase	Collado et al., 2012 [35]
	h. Interferon gamma-induced protein 10 (IP-10)	None reported	-
	i. Monocyte chemoattractant protein-1 (MCP-1)	None reported	-
	j. Interferon Gamma (IFN-γ)	Increase (colostrum) Decrease (1-month milk)	Collado et al., 2012 [35]
	k. Transforming growth factor beta (TGF-β)	Decrease	Collado et al., 2012 [35]
	l. Tumor necrosis factor-alpha (TNF-α)	No alteration	Fields et al., 2017 [18]
		Increase (colostrum) Decrease (1-month milk)	Collado et al., 2012 [35]
5.	Cytokines inhibitors		
	a. Tumor necrosis factor receptor-I (TNFR I)	None reported	-
	b. Tumor necrosis factor receptor-II (TNFR II)	None reported	-
6.	Growth Factors		
	a. Epidermal growth factor (EGF)	Decrease	Khodabakhshi et al., 2015 [19]
	b. Heparin-binding EGF-like growth factor (HB-EGF)	None reported	-
	c. Insulin-like growth factor 1 (IGF-1)	Increase	Khodabakhshi et al., 2015 [19]
	d. Nerve growth factor (NGF)	None reported	-
	e. Vascular endothelial growth factor (VEGF)	None reported	-
7.	Hormones		
	a. Adiponectin	Increase	Martin et al., 2006 [36]
	b. Calcitonin	None reported	-
	c. Erythropoietin (Epo)	None reported	-
	d. Insulin	Increase	Fields et al., 2017 [18]
	e. Ghrelin	Decrease	Khodabakhshi et al., 2015 [19]; Zhang N, et al., 2011 [37]
	f. Leptin	Increase	De Luca et al., 2016 [9]; Lemas et al., 2016 [10]; Uysal et al., 2002 [11]; Andreas et al., 2014 [12]; Quinn et al., 2014 [13]; Eilers et al., 2002 [14];
	g. Obestatin	Decrease	Aydin et al., 2008 [38]
	h. Resistin	No alteration	Andreas et al., 2016 [39]; Savino et al., 2012 [40]
	i. Somatostatin	None reported	-
8.	Immunoglobulins		
	a. IgA	Increase (colostrum and serum)	Miranda et al.,1983 [41]; Islam et al., 2006 [33]; Fujimori et al., 2015 [42]
	b. IgG	Increase	Miranda et al., 1983 [41]
		Decrease	Islam et al., 2006 [33]
		Constant (colostrum) Decrease (serum)	Fujimori et al., 2015 [42]
	c. IgM	Increase	Islam et al., 2006 [33]
		Increase (colostrum) Decrease (serum)	Fujimori et al., 2015 [42]
	d. Insulin-like-growth factor-binding proteins (IGFBP)	Decrease	Khodabakhshi et al., 2015 [19]
9	Lipids	Increase (triglyceride and cholesterol)	Fujimori et al., 2015 [42]
		Increase (saturated fatty acids) Decrease (n-3 fatty acidss) Decrease (unsaturated to saturated fatty acid ratio) Increase (ratio of n-6 to n-3)	Makela et al., 2013 [43]
		Increase (ratio of n-6 to n-3) Decrease (lutein, docosahexaenoic acid, eicosapentaenoic acid, and docasapentaenoic acid)	Panagos et al., 2016 [15]
10.	Microbiota	*Bifidobacterium* (decrease) *Staphylococcus* (increase)	Delzenne & Cani, 2011 [44]; Collado et al., 2012 [35]
11.	Nucleic Acids	None reported	-
12.	Oligosaccharide and Glycans		
	a. Human milk oligosaccharides (HMOs)	No alteration	Azad et al. 2018 [45]
	b. Gangliosides	None reported	-
	c. Glycosaminoglycans (GAGs)	Decrease	Cerdo et al., 2018 [46]
	d. Osteoprotegerin	None reported	-
	e. Soluble CD14s (SCD14s)	Decrease	Collado et al., 2012 [35]
13.	Other Proteins		
	a. Alpha-Lactalbumin (LALBA)	Increase (6–15 days postpartum) Decrease (16–30 days postpartum) Decrease (31–60 days postpartum) Increase (>60 days postpartum)	Sanchez-Poso et al., 1987 [47]
	b. Alpha-1 Antitrypsin (AAT)	None reported	-
	c. Alpha-Amylase (α-amylase)	None reported	-
	d. Bile Salt Stimulated Lipase (BSSL)	None reported	-
	e. Casein	Decrease	Jevitt et al., 2007 [48]
	f. C-Reactive Protein (CRP)	Increase	Whitaker, 2017 [8]
	f. Folate-Binding Protein (FBP)	None reported	-
	g. Haptocorrin	None reported	-

**Table 2 nutrients-11-01284-t002:** Alteration of immunological properties by other influential factors.

Influence Factors	Immunological Properties	Alteration
Malnourishment	Lysozyme	A study by Hennart et al. (1991) demonstrated that malnourished mothers had up to four times higher concentration of lysozyme than well-nourished mothers [199].
	Lactoperoxidase	A study by Chang (1990) showed that the concentration of lactoperoxidase in human milk of malnourished Chinese women decreased up to 50% [200].
	sIgA	The concentration of sIgA in milk of malnourished mothers was lower than the concentration of sIgA in milk of normal weight mothers [41].
	IgG	A study by Miranda et al. (1983), with malnourished Colombian mothers, demonstrated lower IgG concentrations compared to milk of well-nourished mothers [41].
	Alpha-lactalbumin	A study by Lonnerdal et al. (1976) in Ethiopian and Swedish mothers demonstrated that alpha-lactalbumin concentration in well-nourished mothers was higher than in milk of malnourished mothers [178].
	Lymphocyte	The mean quantity of lymphocyte plasma cells in colostrum of mothers with low BMI mothers was higher than that of mothers with normal BMI [33].
Lactation Stages	Macrophages	The composition of colostrum is high in macrophages (30%–50% in leukocytes) [201]. However, macrophage concentration was found to decrease after 1 month of lactation and maturation of human milk [22,202].
	G-CSF	A study by Calhoun et al. (2000) demonstrated that the highest concentration of G-CSF in human milk was 1–2 days postpartum. MIF concentration is highest at the first month of lactation and continues to decrease with time of lactation [203].
	IL-6	The IL-6 levels were at similar levels in milk samples taken at 1 month and 3 months postpartum [8].
	IL-7	A showed that mean concentration of IL-7 was influenced by the age of infants. IL-7 concentration in human milk of 6 months breastfed infants (103.5 ± 37.8 pg/dL) was higher than IL-7 concentration of 2 months of breastfed infants (69.8 ± 40 pg/dL) [204]. In a study of healthy mothers’ milk over time, the concentration of TNFα in colostrum was higher than the concentration in transitional and mature milk [205]. Another study showed that the TNFα concentration in colostrum (402.80 ± 29.65 pg/mL) and mature milk (178.30 ± 14.41 pg/mL) were higher than the TNFα concentration in transitional milk (135.50 ± 8.26 pg/mL) [91].
	EGF	In milk of 33 mothers at day 1–7 postpartum, the concentration of EGF was reported to range from 33.3 to 184.3 (median 71.2) ng/mL [107].
	IGF-1	A study by Erikkson et al. (1993) showed that the concentration of IGF-1 in the colostrum to transitional milk (decreased five-fold day 1 to day 3, and then remained relatively constant through day 8 of lactation [206].
	NGF	A study by Ai et al. (2012) measured NGF levels of seven healthy breastfeeding mothers on the first 3 days postpartum. Measurement of NGF levels showed very high variation and NGF levels postpartum milk were, 236 ± 332 ng/L, 173 ± 113 ng/L, and 178 ± 248 ng/L, for 1, 2 and 3 days, respectively, but were not significantly different over time [207].
	VEGF	A study by Kobata et el. (2008) demonstrated decreasing concentrations of VEGF from 1 to 7 days postpartum [107].
	Ghrelin	While the ghrelin concentration was high in breast milk, there were significant increases in infants’ weight gain at 4 months of age [208]. The negative correlation between ghrelin concentration in serum and infants BMI occurred at the first month of life [146].
		Another study also showed that low concentration of ghrelin in infants would slow the weight gain during the first 3 months [209]. Transitional milk and colostrum contain small amounts of Immunoglobulin G (IgG) while the amount of IgG is higher in mature milk [20].
	Lipid	Colostrum, transition milk, and mature milk contains lipid concentrations of 3–4, 7.2, and 56.2 g/L respectively [210].
	Gangliosides	Around 6%–10% of total lipid mass in human infant brain consists of gangliosides. This concentration will be increase up to 3-fold from 10-weeks’ gestation to 5 years of age [211]. GM1, GM3, and GD3 are three types of gangliosides in human milk that bind pathogens without causing inflammatory reactions [212]. A study by Thakkar et al. (2013) demonstrated that the concentration of gangliosides in human milk was significantly higher at 120 days postpartum in male infants [213].
	α1-antitrypsin	The concentration of α1-antitrypsin from 190 human milk samples of 94 maternal mothers between 1 and 160 days after delivery was found to decrease over time [179].
	Alpha-amylase	The decrease in alpha-amylase concentration could range up to 35% (*p* < 0.001) during the first 3 months [214].
Pregnancy length	Chemokines	A study by Michie et al. (1998) showed that there were no differences in chemokines concentration between human milk of mothers that delivered preterm or full term [215].
	G-CSF	The production of G-CSF (at the first 2 days of colostrum) of premature delivered mothers was lower than that of term mothers [216].
	TNFα	A study demonstrated that the concentration of TNFα in colostrum of mothers with premature babies was lower than the mothers with full term babies [205].
	Alpha-amylase	No differences in alpha-amylase concentration were found in the human milk produced for preterm infants or full-term infants [217].
	BSSL	There was no significant difference of characteristics of BSSL found in the human milk produced from mothers with preterm or full-term infants [218].
	Osteoprotegerin	The concentration of osteoprotegerin in the milk of mothers whose deliver their infant prematurely and full term was not significantly different [172].
Type of Breastfeeding	IL-7	A higher concentration of IL-7 was found in exclusively breastfed infants compared to those that received mixed feeding [204].
Nutritional Status of Infant	IGF-1	A study showed that high BMI of infant was negatively correlated with the concentration of IGF-1 in human milk [19].

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
