# Peer review of "The Function and Alteration of Immunological Properties in Human Milk of Obese Mothers"

_nutrients, 2019, doi:10.3390/nu11061284_

Round 1

Reviewer 1 Report

The authors of the current review have performed an extensive literature review on the impact of maternal obesity on the concentration of immunological properties in human milk. Review covers altogether 195 references, however it concludes that this topic needs further investigation as only 28 studies have reported findings related to this extensive topic. Overall, even though the topic is well defined, the main aim of the review is not always clear. This might be due to the lacking evidence in the field. Parts of the paper need major review keeping in mind the main aim of the study – the impact of maternal obesity on the immunological properties of human milk.

Main main concern is that even though the table 1 shows that there are many components of human milk with no reported findings regarding effects of maternal obesity, the authors have still used quite a lot of time and effort to summarise f. ex. the findings regarding the malnoursihment of mothers or other aspects not always relevat regarding the aim of the review. I would suggest, to increase the readibility of the paper, that the authors focus in this review mainly on the components with published findings related to the effects of maternal obesity. The authors might want to add separate discussion section where they could discuss the findings in relation to the findings related to malnourishment or others factors and leave the main parts of the review to the findings describing the effect of maternal obesity.

A few minor comments:

Would it be more informative regarding the topic of the review to state in the beginning of introduction the percentage of mothers with weight above the range of normal weight than the decreasing percentage of mothers with normal body weight?

Line 26 Sentence describing the increased leptin concentration, the reference is missing.

The composition of human milk changes quite a lot during the course of lactation. I would suggest shortly describing the different phases of the lactation and adding f ex the definition of colostrum in the introduction as it is mentioned later in the review.

Line 55 reference missing

Sentence "Certain studies ..." also needs reference.

Line 74 could you interpret Weight for Height in terms of BMI?

The conclusion is stated as describing the nutritional status of mothers, not the obesity of mothers.

The practice points stated do not follow the main aim of the review.

The authors have done extensive work - the reporting of the findings just needs a lot of revision in regard of the aim of the paper.

Author Response

1st Reviewer Response

Responses

Reviewer’s Comments

The authors of the   current review have performed an extensive literature review on the impact of   maternal obesity on the concentration of immunological properties in human   milk. Review covers altogether 195 references, however it concludes that this   topic needs further investigation as only 28 studies have reported findings   related to this extensive topic. Overall, even though the topic is well   defined, the main aim of the review is not always clear. This might be due to   the lacking evidence in the field. Parts of the paper need major review keeping in mind the main aim   of the study – the impact of maternal obesity on the immunological properties   of human milk.

Response 1

We have revised the aim of the study to more clear specifying   that the primary aim examined maternal overweight obesity and immunological   properties of human milk. Additionally, we removed the alterations due to   other factors and placed them in a new Table 2.

Main concern is that   even though the table 1 shows that there are many components of human milk   with no reported findings regarding effects of maternal obesity, the authors   have still used quite a lot of time and effort to summarise f. ex. the   findings regarding the malnoursihment of mothers or other aspects not always   relevat regarding the aim of the review. I would suggest, to increase the   readibility of the paper, that the authors focus in this review mainly on the   components with published findings related to the effects of maternal   obesity. The authors might want to add   separate discussion section where they could discuss the findings in relation   to the findings related to malnourishment or others factors and leave the   main parts of the review to the findings describing the effect of maternal   obesity. 

Response 2

We separated the obesity and other influence factors of   immunological properties level in human milk. The table 2 contains the other   influence factors of immunological properties. Therefore, the main part of   discussion is focused on the influence of maternal obesity to the   immunological properties in the human milk.

Would it be more   informative regarding the topic of the review to state in the beginning of   introduction the percentage of mothers with weight above the range of normal   weight than the decreasing percentage of mothers with normal body weight?

Response 3

We have added a sentence that describes the percentage of   pregnant mothers with overweight and obesity worldwide and in the US.

Line 26 Sentence   describing the increased leptin concentration, the reference is missing.

Response 4

The references have been added. The number of references is 9-14.  

The composition of   human milk changes quite a lot during the course of lactation. I would   suggest shortly describing the different phases of the lactation and adding f   ex the definition of colostrum in the introduction as it is mentioned later   in the review.

Response 5

We added a description about colostrum, transitional milk, and mature   milk to the article.

Line 55 reference   missing

Response 6

We have completed the reference. The new reference number is 28.

Sentence "Certain   studies ..." also needs reference. 

Response 7

We have completed the reference. The reference number is 29.

Line 74 could you   interpret Weight for Height in terms of BMI?

Response 8

We added an explanation in the article that weight for height   was an older terminology that has been replaced by BMI over 30 for obesity   today.

The conclusion is   stated as describing the nutritional status of mothers, not the obesity of   mothers.

Response 9

We have changed the nutritional status of mothers with obesity   and revised the sentence to make it clearer.                                                                                                                                                                                                                                                                                                                                                    

The practice points   stated do not follow the main aim of the review. 

Response 10

The main aim of this review is to demonstrate the immunological   functions of specific properties in human milk and the alterations that occur   in conditions of overweight and obesity mothers. While 29 properties were   found to change, the totality of these changes on the long term health of the   infant is not known. The practice points should help medical practitioner,   and other health professionals, prevent the alteration of immunological   properties in human milk prior to giving birth and after childbirth, through   monitoring weight gain, while still encouraging breastfeeding due to well   established benefits to the mother (energy balance) and infant (overall   health).

Reviewer 2 Report

INTRODUCTION

The DARLING (Davis Area Research on Lactation, Infant Nutrition and Growth) study showed important determinants of human milk composition to include maternal ideal body weight (%IBW), protein intake, nursing frequency,

menstruation, and parity (9).” The DARLING study also reported that during rapid weaning, when milk volume falls below 300 mL/day, the protein content increases to 20%, it is worth to be included in this introduction.

2.2 Cells.

Stem cells. Recent data support high efficiency for nasal application of stem cells from breast milk for rescuing neonatal brain injury (Keller, Eur J Pediatr 2019). Worth to be mentioned.

2.5 Growth factors.

I would discuss the role breast milk IGF-1 to prevent retinopathy among preterm infants.

2.6 Hormones

Insulin. Offspring born to women with pregnancies complicated by diabetes are at increased childhood risk of developing impaired glucose tolerance. It correlates with volume of diabetic breast. I would add this topic to this paragraph.

Author Response

Dear Reviewer 2,

Please see our responses in the attached file.

Thank you very much.

Kind regards,

Ummu D Erliana & Alyce D Fly

Round 2

Reviewer 1 Report

The authors have successfully done extensive revisions for this manuscript and I believe it has been very much improved. Congratulations! 

The meaning of obesity versus other influencing factors is now more clearly stated and I think the Table 2 is a very good summarisation of the current knowledge in the field. 

Minor text editing is still needed.